GenomePeek—an online tool for prokaryotic genome and metagenome analysis

McNair Katelyn 1 2 deprekate@gmail.com
Edwards Robert A. 1 2 3 4
1 Department of Computer Science, San Diego State University , San Diego, CA , USA
2 Department of Biology, San Diego State University , San Diego, CA , USA
3 Computational Sciences Research Center, San Diego State University , San Diego, CA , USA
4 Mathematics and Computer Science Division, Argonne National Laboratory , Argonne, IL , USA
Scaria Vinod
Electronic publication date: 2015 Jun 16
Publication date: 2015
Volume: 3
Electronic Location ID: e1025
Received 2014 Oct 4; Accepted 2015 May 25
Copyright: © 2015 McNair and Edwards
Copyright year: 2015
Copyright holder: McNair and Edwards
License: This is an open access article distributed under the terms of the Creative Commons Attribution License, which permits unrestricted use, distribution, reproduction and adaptation in any medium and for any purpose provided that it is properly attributed. For attribution, the original author(s), title, publication source (PeerJ) and either DOI or URL of the article must be cited.
License URL: https://creativecommons.org/licenses/by/4.0/

Keywords: Genome, Metagenome, Taxonomic, Bacteria, Sequencing, Population, Distribution, Archaea, Abundance

Funding: National Science Foundation DBI-0850356 MCB-1330800 DEB-1046413 Information and Intelligent Systems Division of the National Science Foundation CNS-1305112 This work was supported by grants from the National Science Foundation [DBI-0850356, MCB-1330800, and DEB-1046413]. High performance computation was provided by award CNS-1305112 from the Information and Intelligent Systems Division of the National Science Foundation. The funders had no role in study design, data collection and analysis, decision to publish, or preparation of the manuscript.

==============================
As more and more prokaryotic sequencing takes place, a method to quickly and accurately analyze this data is needed. Previous tools are mainly designed for metagenomic analysis and have limitations; such as long runtimes and significant false positive error rates. The online tool GenomePeek (edwards.sdsu.edu/GenomePeek) was developed to analyze both single genome and metagenome sequencing files, quickly and with low error rates. GenomePeek uses a sequence assembly approach where reads to a set of conserved genes are extracted, assembled and then aligned against the highly specific reference database. GenomePeek was found to be faster than traditional approaches while still keeping error rates low, as well as offering unique data visualization options.

Introduction

With the cost of sequencing falling, microbial genomes are being sequenced at an increasing rate. Currently there are over 2,000 completed prokaryotic genomes in NCBI (Benson et al., 2009; Sayers et al., 2009), almost 15,000 prokaryote genomes in the SEED database (Overbeek, Disz & Stevens, 2004) and about 75,000 more that are unassembled in the Sequence Read Archive. There are also about 35,000 metagenomes in NCBI and about 90,000 metagenomes available from MG-RAST (Meyer et al., 2008). While complete genome sequencing gives us detailed knowledge about a single prokaryotic species, metagenomic sequencing gives us a broad overview of the microbial environment (Dinsdale et al., 2008). Whether analyzing genomic or metagenomic sequencing, one of the main goals is to identify the taxonomic origin of the specie or species present (Belda-Ferre et al., 2012; Mande, Mohammed & Ghosh, 2012; Carr, Shen-Orr & Borenstein, 2013; Silva et al., 2014).

There are two typical approaches to identifying the species present in a metagenome. The most common methods use homology searches against a reference database of known taxonomic lineage (Altschul et al., 1997; Meyer et al., 2008; Segata et al., 2012). In contrast, ensemble approaches use signature data from all of the reads, such as protein domain frequencies or k-mer composition (Meinicke, Aßhauer & Lingner, 2011; Silva et al., 2014). Homology-based methods generally use protein level alignments due to the highly divergent and mutable nature of prokaryotic genomes. The problem with this approach is that metagenomic sequencing reads tend to be relatively short when compared to protein open reading frames. The average length of prokaryotic genes that encode proteins is about 750bp (Brocchieri & Karlin, 2005). The current sequencing technologies produce reads with an average length of between 30 and 700bp (Buermans & den Dunnen, 2014). Regardless of the technology used, sequencing data will contain fragments of open reading frames because of the stochastic nature of the sampling. The resulting predicted proteins are a mix of partial proteins and partial erroneous translation of non-protein coding regions. The shorter the protein fragment the harder it is to identify the top hit to that fragment, because of shared homology across prokaryotes (Wommack, Bhavsar & Ravel, 2008). One of the first genes used for taxonomic identification of prokaryotes, because it is universally present in all prokaryotes and highly conserved, encodes the small subunit 16S ribosomal RNA (Woese & Fox, 1977; Lane et al., 1985). One of the problems with 16S based analyses is that because the number of copies of the 16S gene in a genome varies from one to as many as fifteen copies per genome, abundance measures must be normalized based on the copy number per genome before 16S sequences can be used as a quantitative measure (Angly et al., 2014). Consequently, other genes have been proposed to be useful as an alternative for phylogenetic identification of the organisms in a sample, including recA, rpoB, groEL, sodA, gyrB, nifD, fusA, and dnaJ (Holmes, Nevin & Lovley, 2004; Adékambi & Drancourt, 2004; Ghebremedhin et al., 2008; Weng et al., 2009).

In contrast to metagenomic samples, identifying the taxonomy of genomic data is usually performed after assembling the reads into contigs, but is also based on, and is influenced by, a priori knowledge of the organism that was cultured for sequencing. The assembly of reads into contigs allows complete open reading frames to be identified. These complete genes can then be used to identify the taxonomy of the strain that was sequenced. Tools such as RAST (Aziz et al., 2008; Overbeek et al., 2013) provide a list of similar organisms based on cumulative similarities based on BLASTP searches. The identification of the species present in the genome, and indeed the assembly and downstream annotation, are hindered if the initial culture was not pure. Impure cultures can arise from poor microbiological techniques, but in our studies of environmental organisms, we have found several isolates that always contain multiple organisms. We suspect that these organisms form a tight mutualistic relationship and hence their continued co-culturing (M Doane & EA Dinsdale, 2014, unpublished data).

To overcome the limitations and problems associated with the current methodology for analyzing metagenomic data and to implement a tool for pre-screening genomic sequencing data; we developed the web-based tool GenomePeek (available at https://edwards.sdsu.edu/GenomePeek). GenomePeek quickly identifies the prokaryotic species present in a set of sequencing reads by finding all reads in sequencing data that are homologous to a set of highly conserved genes useful in distinguishing prokaryotic taxonomy; assembles those reads into contigs, and then uses complete open reading frames to determine the phylogeny of each assembled gene. GenomePeek currently analyzes four prokaryotic genes: 16S, recA, rpoB, and groEL, and four eukaryotic genes (18S, RAD51, HSP60, RPB2). When genomic sequencing data is analyzed with GenomePeek the identity and purity of the culture is measured, and when metagenomic sequencing data is analyzed the taxonomic distribution of the environment is measured.

Materials & Methods

Algorithm

Four databases corresponding to genes: 16S, recA, rpoB, and groEL were created. The 16S database was downloaded from NCBI and contained 9,254 nucleic acid sequences. The three other sets were curated by downloading all full-length amino acid sequences from UniProt (Apweiler et al., 2004), and the source nucleic acid sequences from ENA (Leinonen et al., 2010). For each database duplicate and erroneous sequences were removed. For species where rpoB is split into two smaller genes, B′ and B″, the two sequences were concatenated. The three protein sets contain 6,668, 6,826, and 6,884 sequences respectively. The analogous human sequences genes 18S, RAD51, HSP60, and RPB2, were added to the reference databases to screen for human contamination (Scott, 1973; Sweetser, Nonet & Young, 1987; Venner & Gupta, 1990; Shinohara, Ogawa & Ogawa, 1992). To decrease runtime, CD-hit (Li & Godzik, 2006) was used to cluster the sequences that had 90% or more similarity, and the exclusion of those sequences created a smaller non-redundant databases for each of the four genes. The BLAT program (Kent, 2002) is used to query the user-provided input sequences against each of the four smaller databases created by CD-hit. Only reads that have an E-value below 10−5 and greater than 80% sequence identity are included in the assembly. Assembly is then performed with the CAP3 (Huang & Madan, 1999) program using default values except that the overlap is shortened to 20bp. Contigs assembled by CAP3, and the remaining singlet sequences that were not assigned to a contig are combined into a query file. To determine the taxonomy of a contig/singlet, the NCBI algorithm BLAST (Altschul et al., 1997) is used to search the appropriate redundant database and to calculate the bit-score of all hits with an E-value less than 10−10. The MEGABLAST algorithm from BLAST+ version 2.2.29 (Camacho et al., 2009) is used for 16S and 18S sequence comparison, while the BLASTX algorithm from the blastall (Altschul et al., 1997) suite is used to search for the protein sequences. The blastall suite is used as version 2.2.26 was the last BLASTX program to feature frame shift error correction, which is critical for individual reads that may be found in a metagenome. A contig/singlet is assigned the genus and species of the hit with the highest bit-score from BLASTX. If there are ambiguous hits from BLASTX searches, a subsequent MEGABLAST search is performed and genera not in the top result are excluded. If there are still multiple ambiguous hits with the highest bit-score, then only the genera of those hits are displayed in the plot, however the full data is available for download. Abundance values are calculated by multiplying the length of the alignment by the number of reads that went into that contig/singlet.

Interface

The web interface and background processing programs are written in PHP and implement the Blue Imp JQuery package (http://jquery.com/) to facilitate data submission and retrieval. The PHPlot package (http://phplot.sourceforge.net/) is used to generate the pie graphs. The C source code of Seqtk (https://github.com/lh3/seqtk) was modified and is used to quickly parse and extract sequence data from fasta and fastq input files, and the program faSomeRecords (https://github.com/adamlabadorf/ucsc_tools) is used to parse quality files.

Data

Three popular sequence analysis tools were compared to GenomePeek. The computers used for testing were the same that GenomePeek runs on, which is a 42 node cluster. Each node runs CentOS 6.3; has 8 Intel Xeon 2.6 GHz processors and 128 GB of RAM. With the exception of MG-RAST, all timing tests were run on private dedicated nodes, with four threads. A flowchart of the pipeline of GenomePeek is visualized in Fig. 1.

Figure 1 A flowchart of the pipeline of GenomePeek.

1. Blast+ version 2.2.29 (Camacho et al., 2009) and the November 7, 2013 build of the NT database were database were downloaded and installed on the cluster. For BLAST analysis, default parameters were used. The hit with the highest bit score was assigned as the correct hit, and in cases where there was more than one hit with the highest bit score, the read value was divided across the top hits.

2. MetaPhlAn version 1.7.7 (Segata et al., 2012) was also run with default parameters, using Bowtie2 (Langmead & Salzberg, 2012) as its search program. To perform well on longer read datasets, such as the FAMES, Bowtie2 would need to be run with the local alignment option.

3. The MG-RAST website 3.3.7.3 (Meyer et al., 2008) was used with all default parameters with the exception of percent identity of the translated reads versus the M5NR protein database. To determine the best percent identity cutoff, the RefSeq (Pruitt, Tatusova & Maglott, 2005) annotated reads for the FAMES databases (Mavromatis et al., 2007) were downloaded using the API. The percent identity cutoff was varied from 60% to 100%, and the reads were parsed to determine if the predicted taxonomy of the read was incorrectly identified (False Positive) or no taxonomy was predicted (False Negative). The errors were individually summed for each percent identity cutoff and plotted (Fig. S1); in addition the total error was calculated by summing both types of error. The minimum for total error was 93% and so all subsequent MG-RAST data analysis was performed using a 93% identity cutoff value. Runtimes for MG-RAST were calculated by finding the time difference between when the uploaded sequence file was compressed and when the results file was compressed.

Artificial genome sequencing files of the ten species that have the most sequencing files in the NCBI SRA (Benson et al., 2009; Sayers et al., 2009) were constructed using Grinder (Angly et al., 2012). The species are: Campylobacter jejuni, Clostridium difficile, Escherichia coli, Mycobacterium tuberculosis, Neisseria meningitidis, Salmonella enterica, Staphylococcus aureus, Streptococcus pneumoniae, Streptococcus pyogenes, andVibrio cholerae. The full genome sequence and plasmids for each strain was downloaded from NCBI, and the accession numbers were, respectively: PRJNA57587, PRJNA41017, PRJNA57781, PRJNA58417, PRJNA57817, PRJNA57799, PRJNA57903, PRJNA57857, PRJNA57845, and PRJNA57623. The artificial sequencing files were modeled to represent Illumina files: one million reads of 101bp long, with a 4th degree polynomial mutation distribution, and 4 substitutions for every insertion/deletion. In addition ten real sequencing files were downloaded from the SRA that matched these strains and were Illumina sequencing. The accession numbers and average read lengths were, respectively: ERR162944 (101bp), SRR593195 (101bp), SRR587217 (231bp), SRR974839 (101bp), ERR330007 (145bp), SRR1060692 (100bp), SRR592258 (100bp), ERR330014 (142bp), ERR104779 (100bp), and ERR351211 (105bp).

Artificial contaminated sequencing files were created using both simulated reads created by Grinder from complete genomes: S. pneumoniae (PRJNA57857) and S. pyogenes (PRJNA57845), or randomly selecting reads from real sequencing S. pneumoniae (ERR330014) and S. pyogenes (ERR104779) files. Reads from ERR330014 were trimmed to 100bp from the beginning to match the read length of ERR104779. A mixture of S. pyogenes contaminated with low levels of S. pneumoniae was selected, due to the fact that these two are related at the Genus level, and thus make it slightly more difficult for taxonomic assignment. We chose S. pyogenes as the dominant species since the real sequencing file had more reads which allows us to create larger simulated contaminated sequencing files. For the contaminated sequencing files constructed from simulated reads, Grinder was used as before, and reads were simulated from the two different complete genome records, PRJNA57857, and PRJNA57845, at varying abundances. For the contaminated sequencing files constructed from real reads, seqtk was used to randomly select reads from the real sequencing files ERR330014, and ERR104779, at varying abundances. Reads from ERR330014, were trimmed 100bp from the start to match the read length of ERR104779. Each of the 50 artificial contaminated sequencing files contained one million reads; with the abundance of S. pneumoniae reads ranging from zero to five percent in 1% increments, and then 5% increments up until the metagenome consisted of 100% S. pneumoniae.

Three different sets of metagenomic data were analyzed. The first set of artificial metagenome sequencing files were downloaded from the FAMES website (Mavromatis et al., 2007). The FAMES metagenomes were designed to model the complexity and phylogenetic composition of real metagenomes and were by created by combining random reads from 113 isolate genomes. There are three FAMES metagenomes that correspond to three different phylogenic distributions: low, medium and high complexity. Of note is that while the MetaPhlAn reference database had at least one representative for each genus in the FAMES metagenomes, there were, however, twelve species in the FAMES metagenomes that were not in the MetaPhlAn database. We considered whether to modify the FAMES metagenomes by removing reads from those twelve species, however tailoring test data to suit a specific program, which was released five years after the FAMES metagenomes and has not been updated in two years, also did not seem fair, so we left the FAMES metagenomes unaltered. The second set consisted of the four Human Microbiome Project mock community metagenomes (The Human Microbiome Project Consortium, 2012). They are constructed from 21 known organisms, and are in either an “even” or a “staggered” abundance. They were subsequently sequenced with both 454 and Illumina technology. The last set, 9MM, was two artificial metagenomes constructed from merging nine diverse bacterial and eukaryotic species, then the mixture subjected to next generation sequencing to obtain experimental metagenomic sequences (Tanca et al., 2013). Metadata such as read count and average read length can be found in Table S6.

Results and Discussion

Genomic taxonomic screening

Previously, the only public tools available to screen genomic sequencing data for contamination generally just screened for model organisms, RNA, or entire Domains (Schmieder & Edwards, 2011). The most common contaminant is human DNA, and has been shown to be present in 22.39% of the non-primate trace archives (Longo, O’Neill & O’Neill, 2011), however one area often overlooked is contamination from other prokaryotes. Such contamination can arise from incomplete isolation of a bacterial/archaeal strain, the contamination of another strain to a pure culture, or a mutualistic relationship between two organisms. When contaminated sequencing data is assembled the results are a chimeric amalgamation of the two genomes. Only manual annotation and curating, after assembly, would identify the “genome” as chimeric, if it were discovered at all. With GenomePeek a researcher can screen their genome sequencing files for integrity in a matter of minutes. Although tools designed for metagenomic analysis can be used on genomic sequencing data, most of those tools have significant false positive rates when classifying at the species level, which would make analysis with these tools difficult. Metagenomics analysis tools were not typically designed for genomic screening; we analyzed ten different artificially constructed genome files using GenomePeek and three of the most popular sequence analysis tools (Fig. 2).

Four different sequence analysis software packages were used to determine the phylogeny of the reads in simulated read sequencing files and the real sequencing files of the ten species that have the most sequencing files in the NCBI SRA (Benson et al., 2009; Sayers et al., 2009) (Fig. 2). GenomePeek had the lowest error rate for every species as well as the lowest combined average error rate. The averaged error rates for GenomePeek, MG-RAST, MetaPhlAn, and MEGABLAST were: 1.36%, 22.61%, 0.00%, and 4.23% respectively (Table S1). Only the M. tuberculosis genome produced any significant error with GenomePeek, this is because M. tuberculosis has two copies of groEL that are identical at the nucleotide level to groEL from M. bovis and M. africanus. However this ambiguous assignment would be resolved after a visual inspection of the GenomePeek GUI, since the other three genes all unambiguously hit to M. tuberculosis. This identical nucleotide similarity also caused high error rates for M. tuberculosis in all of the analysis tools. For MG-RAST the highest annotation error was for E. coli, which has a very high similarity not only to other Escherichia species but also the genus Shigella. MetaPhlAn showed zero false positive error and ran exceptionally fast, however MetaPhlAn only has a database of 1,221 different prokaryotic species, and since it, as well as MEGABLAST, relies on nucleotide similarity, divergent species could be completely overlooked.

Figure 2 False positive error when using four different programs to analyze complete genome sequencing files.

Resulting error when analyzing either artificially constructed or real genome sequence files. The false positive error is calculated from the percentage of sequences that do not hit to the originating species.

To determine at what level GenomePeek can detect contamination, we analyzed two sets of artificial data where S. pyogenes has been contaminated with S. pneumoniae. The level of contamination at which two species are distinguishable from the background false positive error, is the most important characteristic, and the abundance of a contaminating species would have to be above the false positive error rate to be observed. We found that for S. pyogenes, GenomePeek had a 0.07% average false positive error for simulated and real reads (Table S2). MG-RAST, MetaPhlAn, and MEGABLAST had 9.26%, 0.00%, and 4.66% false positive error rates respectively. Thus only GenomePeek and MetaPhlAn are able to accurately identify low levels of contamination. When analyzing the contaminated data with GenomePeek we found an average false positive error of 0.52% for files constructed from simulated reads, and 1.67% for files constructed from real reads (Table S2). Thus a positive S. pneumoniae signature became noticeable when S. pneumoniae reached at least 2% abundance (Fig. 3). As the abundance of S. pneumoniae increased, the clearer the signal became, since the false positive error rate stayed relatively constant. Of note is that the predicted abundance of S. pneumoniae is less than expected because S. pneumoniae has half as many 16S operons as S. pyogenes. In addition, an abundance bias is caused by the close sequence similarity of the two strains, since during the assembly step reads from the least present species can get recruited into contigs of the more abundant species, due to the weighted algorithms of CAP3. When we only used recA, rpoB and groEL to compare these data sets, the signal from the S. pneumoniae contamination is much clearer (Fig. S2), as well as clearer low contamination levels. Excluding the 16S data from taxonomic distribution calculations removes the bias caused by species with high rRNA copy numbers.

Figure 3 The GenomePeek predicted taxonomic distribution of simulated complete genome sequencing files that have been contaminated with a second species.

Summed false positive error rates of all four key genes, when GenomePeek is used to analyze simulated sequencing files of S. pyogenes that have been contaminated with S. pneumoniae.

Metagenomic taxonomic distribution

To test the ability of GenomePeek to calculate the abundance of species in metagenomic samples, we analyzed nine different artificial metagenomes. The FAMES metagenomes (simLC, simMC, and simHC) correspond to three different phylogenetic distributions: low, medium and high. When it came to identifying only the genera of the FAMES metagenomes, each of the programs has similar false positive error rates (Fig. 4A, Tables S3 and S4). GenomePeek, MEGABLAST and MG-RAST performed comparably while GenomePeek had a higher false negative rate, caused by the much smaller reference database. Although MetaPhlan had a zero false positive, it had an extremely high false negative error rate. In contrast to genus-level assignments, when comparing species-level assignments the error rate for MG-RAST drastically increased, mostly from more false positives (Fig. 4B). The error rate for MEGABLAST against the NT database only slightly increased, and was by far the lowest across all four programs. Of note is that MetaPhlAn could have performed much better had the marker database been more complete.

Figure 4 Error rates of the different sequence alignment tools, when identifying the genera and species of metagenomes. The false positive error rate is calculated from the percentage of sequences that do not.

Error rates of the taxonomic predictions when using four different programs to analyze artificial metagenomes sequencing files.

One of the advantages of the FAMES metagenomes is that the exact underlying population distribution is known, since the reads were taken from old sequencing technology at a specified abundance distribution. The average read length of the FAMES metagenomes is 957bp, much higher than the average length of reads using current sequencing technology (for example, the average length of a read on MG-RAST is 115bp). To determine the accuracy of each of the programs on current sequencing data, we downloaded the only six artificial metagenomes in the SRA. Four were from the Human Microbiome Project, and two were from a separate research project. In both sets, microorganisms were grown separately, then combined just before being sequenced with either 454 or Illumina technology. Since the species are combined and then sequenced we do not know for certain the actual taxonomic species distribution of the reads, so false negative error rates could not be ascertained. The only tools that failed to identify all organisms present were GenomePeek and MetaPhlAn, and only for organism that were estimated to be less than 0.1% of the population. As before MG-RAST has very high false positive rate, especially when identifying at the species level (Fig. 4B). GenomePeek had the next highest false positive error rates, higher than with the previous FAMES data. MEGABLAST had false positive error rates comparable to the FAMES data. MetaPhlAn was the most accurate at identifying the species present with nearly zero false positive error rates, showing just how well MetaPhlAn works when the query organism is in the MetaPhlAn marker database. For these analyses MetaPhlAn took about a minute to run, GenomePeek takes minutes to hours to run, while MEGABLAST and MG-RAST both take hours to days to run (Table S5).

Conclusions

The software presented here provides a rapid, user-friendly approach for examining the species present in genome or metagenome sequences. It is simple to use, needs no installation and requires nothing more than uploading files. It has error rates comparable to other popular sequence analysis tools with lower runtime. Each of the tools presented here have benefits and drawbacks: MG-RAST’s ability to identify divergent species, but having very high false positive rates; MetaPhlAn with its high accuracy and speed, but limited species; and MEGABLAST’s high accuracy, but slow speeds and divergent species limitations. By utilizing specific well-known taxonomic gene markers for assignment and assembly, GenomePeek not only quickly assigns the reads of a genome or metagenome sequencing file, but it also provides the user with their assembled 16S, recA, groEL, and rpoB genes for further analysis using more sensitive tools. GenomePeek is a quick and simple method for preprocessing prokaryotic sequencing data, and since it is online, it does not require the user to install anything or consume any of the user’s computing resources. Since contamination can be quite common, as is the case of human sequence contamination, any researcher would be wise to check the integrity of their sequencing file before devoting time to filtering, trimming, assembling and/or annotating.

Supplemental Information

Figure S1 The effect of varying the percent identity cutoff on the error rates of the predicted taxomonic species when using MG-RAST

Varying the percent identity cutoff on MG-RAST and its effect on the error rate when identifying the species of the FAMES metagenomic reads.

Click here for additional data file.

Figure S2 The GenomePeek predicted taxonomic distribution of simulated complete genome sequencing files, that have been contaminated with a second species, when 16S data is excluded

Summed false positive error rates when GenomePeek is used to analyze sequencing files of S. pyogenes that have been contaminated with low levels of S. pneumoniae, and using only the genes, RecA, RpoB, and groEL.

Click here for additional data file.

Table S1 False positive rates when identifying the species of artificial sequencing data and real sequencing data

Click here for additional data file.

Table S2 The predicted abundances of S. pyogenes and S. pneumoniae in artificial contaminated sequence data

Click here for additional data file.

Table S3 Error rates of classifying the genus of reads of a metagenomes, by various tools

Click here for additional data file.

Table S4 Error rates of classifying the species of reads of a metagenomes, by various tools

Click here for additional data file.

Table S5 Runtimes for the various sequence files across the four different applications used

Click here for additional data file.

Table S6 Metadata for the various sequence files

Click here for additional data file.

We thank Dr. Fangfang Xia for helpful comments on the manuscript.

Additional Information and Declarations

Competing Interests

Author Contributions

Data Deposition

Robert A. Edwards is an employee of the Argonne National Laboratory.

Katelyn McNair conceived and designed the experiments, performed the experiments, analyzed the data, contributed reagents/materials/analysis tools, wrote the paper, prepared figures and/or tables.

Robert A. Edwards contributed reagents/materials/analysis tools, reviewed drafts of the paper, funding.

The following information was supplied regarding the deposition of related data:

Figshare: http://figshare.com/articles/GenomePeek_Data/992688.

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
