# Peer review of "GenomePeek—an online tool for prokaryotic genome and metagenome analysis"

_PeerJ, doi:10.7717/peerj.1025_

## Round 0.1 · original submission · Major Revisions

· Academic Editor

Major Revisions

Thanks for giving me a chance to review your manuscript. The anonymous reviewers have suggested major revisions in the manuscript, the detailed list is provided below. We would like to see all the comments addressed appropriately in the revised version of the manuscript with a detailed response to each of the comments raised by the reviewer. If you would be unwilling to address any of the comments raised by the reviewer, please add a detailed response on why it would be impossible to address the same

Reviewer 1 ·

Basic reporting

OK

Experimental design

OK

Validity of the findings

OK

Additional comments

This work should be be valuable for the metagenomic research field, however, several issues remain unclear:

(1) The authors stated that 16s rRNA based methods are not quantitative, but their methods could not solve this issue either.

(2) The authors stated that GenomePeek has higher false positive (Figure 3), yet no explanation of why (I do not believe small DB size is the whole reason) and how to improve given.

(3) For speed, the authors should check with papers about similar topic (Zhou, Scientific Reports, 2014; Zhou, Genomics, Proteomics and Bioinformatics, 2014, etc.) and possibly compare with their methods.

(4) Some typos: Supplementary Figure 0?

(5) If possible, the authoers might discuss how they could assess the consensus of results based on 4 genes?

Reviewer 2 ·

Basic reporting

Supplementary Figure 1 is incorrectly labelled as "Supplementary Figure 0"

Line 171: The citation for this paper should be edited so as not to imply the existence of a Dr. "T. H. M. P. Consortium".

Figures 1 and 2 would benefit from selecting colors with greater contrast; I might recommend using a dark blue and a light orange.

Experimental design

The description of the execution of GenomePeek indicates that it was run on a cluster with nodes that have multiple processing cores. However, I don't see anything stated in the text regarding whether or not GenomePeek uses multiple cores (or multiple nodes) for its results. This needs to be mentioned, as MetaPhlAn - being run with default options, according to the text - was run with only one processing core, and the comparison between the two programs needs to be a fair one.

Although the average read length is provided for the real sequences for the 10 genomes' read data downloaded from the SRA, read counts are not. The number of reads in each sample has a strong impact on runtime, and needs to be mentioned. (Read counts and mean read lengths should also be reported for the metagenome data.) Supplementary Table 5 might be a good place for this metadata, although please see my comments about the table below.

Validity of the findings

On line 225, the authors state that the false positive rate was "relatively constant" as the abundance of S. pneumoniae increased; however, supplementary table 2 (from which Figure 2 appears to be generated) seems to indicate that the FPR is actually correlated with the abundance of S. pneumoniae, with R = 0.89 for simulated data and R = 0.65 for real data. This would seem to contradict the authors' statement. In addition, the correlation of FPR with "contaminant" abundance makes me concerned about the results shown in Figure 1 - if adding S. pneumoniae to S. pyogenes results in a detectable FPR for the mixture, why do we not see any GenomePeek FPR in Figure 1 for S. pneumoniae alone? The authors should explain (a) the meaning of "relatively constant" in light of the correlation and (b) the lack of FPR for S. pneumoniae in Figure 1 vs. its presence in Figure 2.

This also calls into question the statement in lines 210-211 ("GenomePeek is the most accurate tool for checking the integrity of whole genome sequences."), which appears to be based on the results in Figure 1. I remain unsure of the utility of the single-genome experiments reported on in Figure 1, especially in light of the apparently differing FPR results between them and the contamination experiment. Given that GenomePeek does not analyze the sequences in isolation (i.e., reads are assembled, and may be assembled incorrectly), single-genome experiments only show GenomePeek's lack of false positives, but aren't informative for their purported purpose (to show that GenomePeek can be used to identify the presence of unexpected sequences in a sample - i.e., to "check the integrity" of isolate WGS data). The experiment that would show this, the contamination experiment with the two Streptococcus genomes, yields possibly anomalous results and is not run with all tools. I don't believe the statement in lines 210-211 is supported by the experiments shown; to make this claim, multiple contamination experiments would have to be run, with various combinations of species, and with all of the tools being compared. Furthermore, "accuracy" would have to be defined in a manner that accounts for false negatives as well, as failure to detect contamination certainly impacts accuracy for this use case.

Regarding supplementary table 5, there are a few anomalous results where MetaPhlAn is slower than GenomePeek on the real data or metagenomic data (S. aureus, S. pyogenes, V. cholerae, 9MM-A). These results concern me, as there isn't discussion of them, and they are counterintuitive given the methodologies used by the two programs. (MetaPhlAn uses Bowtie2, while GenomePeek uses BLAT, assembly, and BLAST - intuitively, at least, MetaPhlAn should be faster for all but the smallest data sets.) Could the authors recheck the runtimes on those datasets for those two programs, and if GenomePeek is still faster than MetaPhlAn, provide some explanation for these anomalies?

Depending on the read counts for the real data sets, and the authors' comments on the anomalous results discussed above, I have to take issue with the statement on lines 261-262 ("Both GenomePeek and MetaPhlAn take minutes to hours to run"). A reference to a supplementary table is made, and viewing that table shows that for many of the large data sets, MetaPhlAn's speed is superior - and clearly so - to GenomePeek's. Although the authors' statement is technically correct, it is misleading, as it creates the impression that the differences between the two runtimes are negligible, and this impression is aided by the authors' decision to place the actual runtime comparison in a supplement. This is not to say that I think GenomePeek is in all respects inferior to MetaPhlAn - the use of amino acid sequence search and (hopefully) an up-to-date reference database should provide clear accuracy advantages to GenomePeek over MetaPhlAn. But an accurate and more detailed reflection of the speed comparison should be present in the main manuscript, not in a supplement.

The first half of the final sentence of the conclusion is merely a statement of ambition and hope, and as such can't be supported by any data. (I must also note that hoping for anything to become "standard" in this field is likely to be met with disappointment.) That half of the sentence should be removed or amended.

Additional comments

Several minor typographical issues are present:
In many places, including Lines 155, 212-229, and Figure 2, S. pneumoniae is incorrectly presented as "S. pneumonia".
In line 229, text should read "Data not shown", not "Data not show".
In line 219, text should read "low levels", not "low levers".
In lines 202 and 227-228, gene names and S. pneumoniae should be italicized.
In line 218, a comma should be inserted between "MG-RAST" and "MetaPhlAn".
In line 212, the semicolon should be a comma.
In line 133, no comma should exist in the phrase "plasmids, for each".
In line 161, the word "and" should (probably) be replaced by "that".
In line 191, text should read "these tools", not "these tool"; in addition, I cannot discern the intended meaning of the phrase "... when classifying the at the species level...".
In line 262, "BLAST" should be written as "MEGABLAST" (assuming that is what is being referred to), and text should read "takes hours to days", not "take hours to days".
In line 235, text should read "MEGABLAST", not "MEGBLAST"

Reviewer 3 ·

Basic reporting

No comments.

Experimental design

No comments.

Validity of the findings

No comments.

Additional comments

The manuscript describes a pipeline called GenomePeek that combines previously existing software tools in order to assess the number and type of species/genera present in a sequencing sample according to four established marker genes. The pipeline is accessible via a web interface. The web interface is easy to use and functional. The manuscript claims that GenomePeek "was developed to analyze both single genome and metagenome sequencing files, quickly and with low error rates".

Major Comments:

1) The description of the algorithm is confusing. After reading it several times, it is still unclear to me at which level nucleotide sequences and at which level protein sequences are used. Please provide a figure to illustrate the workflow and the databases.

2) If I search a sequence in a database, and it is known that this sequence actually has a good match in the database, it is no surprise that I am going to find a good match. Some methods may be more sensitive than others, some may be more specific, but it is a feasible problem. I am a bit surprised by your choice of organisms for building artificial (single) genome sequencing files: Why do you choose the ten genomes that have most sequencing files in SRA? Many sequences of those species are in the reference search databases. After reading your manuscript, it is still unclear to me how well GenomePeek works if e.g. the contamination and/or the target genome species is not present in the reference search database on a genus or species (or an even higher ranking) level. Please add experiments that make this clearer.

3) Concerning the simulated and real datasets (for single genomes), I understand that the real datasets have read lengths varying from 101 to 206 nt. However, the current Illumina read length already exceeds those read lengths. I believe your manuscript will benefit from results of tests on actual current read lengths, because users of your tool will most likely submitt state-of-the-art sequencing files that contain longer reads, and it is therefore interesting for those users how well your tool (and the others that you compare to) perform on those read lengths and the Illumina sequencing error profile. Increased accuracy is to be expected but it would be nice to actually see the numbers.

4) I believe that there is demand for a tool such as GenomePeek to quickly estimate whether a sequencing file is contaminated. After reading your results for this purpose, I have the impression that GenomePeek serves this purpose. (Although I raise questions major 2 and major 3.) For metagnomes, I am not convinced that GenomePeek is a very attractive tool because if there is a good database match, MetaPhlAn works better and at feasible speed. I see that GenomePeek can be applied to metagenomes, but I don't see the advantage of doing so in comparison to MetaPhlAn.

Minor Comments:

1) In lines 22/23: "one of the main goals is to identify the taxonomic origin of species or species (...)" sounds confusing. Please rephrase.

2) In lines 33/34: "The current sequencing technologies produce reads with an average length of 100 to 500bp." Please rephrase and make clearer which read length belongs to which technologies, and what's the actual current read length. I assume the 100 nt refer to Illumina sequencing? If yes, the current read length is not 100, anymore.

3) Initially, I wanted to suggest the usage of a rRNA specific database such as arb-silva instead of NCBI. However, I read later, that you previously did use silva, but that one of the other reviewers questioned that database. This is unfortunate. Adding an apropriate reference instead of switching the database might have solved the problem. However, I believe that the effect of the database switch on results of GenomePeek is minor and I now suggest to stick to NCBI - even though arb-silva might still be the more appropriate database for rRNAs.

4) The section "Data" contains not only data, but also previously existing tools that are used to show how well GenomePeek works. A separate section is probably not necessary to make the difference between the two, but dividing into more paragraphs and using enumerations will make this section a lot clearer:

Lines 109-112 describe the computational setting on which you measured runtime.

Lines 112-127 describe the other tools. I suggest you use an enumeration which will make it a lot easier for readers to understand that lines 119-127 refer to the last of three methods.

5) In lines 149-151 and 154-156, you describe that several data sets with varying abundance of the contaminant species were created. Without looking at figure 2 (which is not and should not be referred to at this point in the manuscript), it is impossible to understand which exact data sets were created. Please make this more clear (i.e. stating how many data sets and with which exact percentages of contaminant were created).

6) I am confused by the sentence "The FAMES samples were created with previous sequencing technology in mind and therefore seven other constructed metagenomes were downloaded from the SRA website" in lines 168-170. Which seven metagenomes are those? In the next sentence, you refer to a "second set". Is the second set part of the seven? If so, there are not seven, but only 6 other metagenomes (four from the Human Microbiome Project and two from 9MM).

7) in line 219, correct levers to levels

8) in line 220, correct "... can only be used on with species..."

9) Who do you not show the data supporting your statement in line 229 as a supplementary figure?

External reviews were received for this submission. These reviews were used by the Editor when they made their decision, and can be downloaded below.

---

## Round 0.2 · Major Revisions

· Academic Editor

Major Revisions

The authors are advised to take care of the comments suggested by the reviewers in the manuscript

Reviewer 1 ·

Basic reporting

Actually more experiments have been added, as well as more explanations.

Experimental design

More experiments have been added, which is good for this work.

Validity of the findings

Comparison with other methods should include more recent works that are directly related with processing efficiency, such as the fast Parallel-Meta software using GPU (Su, BMC Systems Biology, 2012).

Additional comments

If some claims are not addressed by the method that the paper reported, then the authors better not to state that or change the tone. For example "One of the problems with using 16S for analysis is that it cannot be used as a quantitative measure".

Also about "The high false positive rate of GenomePeek arises from several reasons. But the main reason is actually due to the small query size": there should be proof for this.

Reviewer 2 ·

Basic reporting

Although the colors for Figure 1 have been changed to a more contrasting pair, Figure 2's colors remain a very similar red/magenta. Please change these to orange and blue (or some other pair that is easily distinguishable - I only recommend orange and blue as they aren't easily confused by people with colorblindness).

I also note multiple requests have been made by other reviewers over the previous submissions to provide a figure to illustrate the general workflow. Although I don't believe such a figure to be an absolute necessity, providing a graphic representation of the workflow certainly should not hurt the ability of readers to comprehend GenomePeek's operation, and would likely aid readers. Given this journal's lack of page limits, I recommend the authors reconsider their position on creating this figure.

Experimental design

From the rebuttal, it appears that at least MetaPhlAn and GenomePeek were run with 4 threads, but it's unclear with how many threads the other programs were run, nor is thread usage mentioned in the manuscript. For example, I note that MetaPhlAn is simply described as being run with defaults, but the times have reduced since the last submission and appear to be 4-thread times, indicating a departure from the defaults. Please explicitly state the thread counts used for all programs in the manuscript.

I also note that the authors used BLAST+ (i.e., MEGABLAST, as no "-task blastn" is present in the metaphlan.py code) for MetaPhlAn instead of Bowtie2 - neither of these is actually a default, as a choice must be made by the user. I would expect Bowtie2 would give similar results to MEGABLAST but be faster, and be the version preferred by most users (I note that MetaPhlAn 2 doesn't appear to even support BLAST usage). Unless the authors can give a compelling reason to the contrary, Bowtie2 should be used with MetaPhlAn here.

Lines 151 to 153 indicate average read lengths that in some cases differ from the average read lengths reported in Supplementary Table 6. For example, a quick check of the SRA reveals that the supplementary table is correct in reporting the E. coli (SRR587217) mean read length to be 231 bp, not the 206 bp reported on line 152. Please check the numbers on these lines and the table to ensure their correctness.

Validity of the findings

The authors now claim "GenomePeek is the best tool... when runtime and divergence are an issue" (line 224). The divergence part of this claim (newly emphasized in this revision) is related to GenomePeek's use of protein-space comparisons vs. other tools' nucleotide-space comparisons. However, no experiments are present in the manuscript that show that GenomePeek gains substantially more sensitivity by using BLASTX as opposed to the other programs. (I note that there is an anecdote about the matter in the rebuttal.) Given MetaPhlAn's strength in terms of speed, I would like to see some sort of quantification of the benefits of using the more computationally-expensive approach of translated alignment. Another reviewer suggested excluding parts of the reference database, which was declined by the authors due to an inability to alter the other tools' databases. I think, however, such an experiment should be done with at least GenomePeek if only to see how much prediction ability is lost when dealing with divergent data. Similar clade-exclusion experiments have been done in other papers, such as the papers for the sequence classifiers PhymmBL (http://www.ncbi.nlm.nih.gov/pubmed/19648916) and Kraken (http://www.ncbi.nlm.nih.gov/pubmed/24580807); the results for Kraken (Table 2) seem to indicate that nucleotide-level matching still has some - although reduced - ability to aid in predictions when dealing with novel species and genera.

"Best" (line 224) is a subjective and inappropriate term when describing GenomePeek's performance across two metrics (runtime and accuracy with divergent data), because GenomePeek is not the best tool on one of the metrics in isolation (runtime) and no standard method exists to combine performance on the two metrics. For example, if I have an enormous amount of divergent data, the runtime advantages of MetaPhlAn may be worth the (potential and unquantified) loss of sensitivity when compared to GenomePeek. The claim must be restated in an objective fashion, in terms clearly backed by evidence presented in the manuscript.

The authors also have added a mention of human contamination in their conclusion (line 295). While human contamination can indeed be common, this mention seems a bit out of place, as it doesn't appear that GenomePeek can detect human or other eukaryotic contamination based on the algorithm the authors presented (if it can, that should be emphasized - the introduction indicates that GenomePeek focuses only on prokaryotic genomes). This mention should probably be removed or discussed in more detail to show its relevance to the paper.

Additional comments

I have several minor comments:
Line 40: "average length of 30 and 700bp" should probably be "average length of between 30 and 700bp"
Line 60: use "BLASTP", not "blastp"
Lines 101 and 102: use "BLASTX", not "blastx", and "BLASTN" or "MEGABLAST" (as appropriate) instead of "blastn"
Lines 157 and 158: genus names can be abbreviated on second mention of the organisms
Line 170: Text should read "zero to five percent", not "zero to five"
Lines 171 and 228: "S. pneumonia" should be "S. pneumoniae".
Line 216: "groEL" should be italicized in its first mention on this line
Line 293: use "prokaryotic", not "procaryotic" - even if the latter is an acceptable alternative spelling, it's not in agreement with the usage in the rest of the paper
Line 296: I recommend using "and/or" instead of "and then", as there are of course different uses for sequence data that don't require assembly or annotation but can benefit from a contamination screen.

Furthermore, three points from my most recent review were claimed to be addressed by the authors in their rebuttal, but have not actually been addressed in the submitted manuscript (I have adjusted line numbers to reflect the current submission):
Figure 2's coloring, as mentioned above.
In line 205, text should read "these tools", not "these tool"; in addition, I cannot discern the intended meaning of the phrase "... when classifying the at the species level...".
In line 255, text should read "MEGABLAST", not "MEGBLAST"

Reviewer 3 ·

Basic reporting

The manuscript has been improved. Authors addressed all issues previously raised, appropriately.

Experimental design

ok

Validity of the findings

ok

---

## Round 0.3 · Major Revisions

· Academic Editor

Major Revisions

The reviewers (especially reviewer 2 ) have suggested extensive correction of the manuscript. In addition, are not convinced on the clear advantages of GenomePeek over other tools. It would be necessary to explain in detail and in an unambiguous way how the software fares better than the others already widely sued by the community.

Reviewer 1 ·

Basic reporting

I still feel that the novelty of the method presented in in manuscript is its weakness.

Experimental design

Reasonable.

Validity of the findings

Results detailed enough but discussions not insightful.

Additional comments

Though this paper present the GenomePeak method that could process genomic and metagenomic data successfully, I think the novelty of the method presented here lack both theoretical advantage and discussions are not insightful.

Reviewer 2 ·

Basic reporting

I disagree with the authors regarding the usefulness of a figure to explain their processing pipeline. Even though, in their rebuttal, they state that the pipeline "can be boiled down to the series INFILE -> BLAT -> ASSEMBLE -> BLASTN -> BLASTX", this statement does not help a reader, as it's not in the manuscript (and it would not be appropriate there). I note the authors have described a figure as "overkill", but yet, three reviewers - on three different revisions of the manuscript - have requested a figure for the pipeline. I don't understand the authors' reluctance to provide a figure here. I've re-read through the paragraph describing the pipeline, and I continue to believe a figure would be beneficial for a reader. The description at present is (quite necessarily) dense with implementation details, and a figure would provide a helpful outline for readers.

Experimental design

With respect to the BLAST/Bowtie2 usage for MetaPhlAn, I imagine the low sensitivity for MetaPhlAn with Bowtie2 on the FAMES data is due to MetaPhlAn's default use of end-to-end alignment, which is likely inappropriate for the Sanger reads used in this experiment. Although MetaPhlAn allows use of another Bowtie2 preset (e.g., "--bt2_ps sensitive-local"), the MetaPhlAn authors state in MetaPhlAn 1.7.7 that local alignment isn't recommended to avoid "overly sensitive hits". I'm not sure if the MetaPhlAn authors intended their program to be run with data from sequencing technology that has had little use in the past 5 years. In any event, if using local alignment solves the sensitivity issue, then simply stating in the manuscript that MetaPhlAn required a non-default local alignment to perform well with Bowtie2 on the FAMES Sanger data would be sufficient for me. (If this is the case, a similar statement should be present in the Figure 3 legend to direct readers to an explanation for MetaPhlAn's strange behavior for that data set.)

Validity of the findings

After the authors removed the statement that GenomePeek was the best when runtime and divergence were an issue, I have no issue with the findings. In response to the authors' note regarding translated search, they may rest assured that I am aware that translated search will likely increase sensitivity; in fact, that is why in a previous review, I stated "the use of amino acid sequence search ... should provide clear accuracy advantages to GenomePeek over MetaPhlAn." Nonetheless, translated search is more expensive computationally than nucleotide-only search. My point of contention was the implicit claim in the now-removed statement that the extra runtime would clearly be worth the increased sensitivity, as the increase in sensitivity was not quantified.

Additional comments

I have the following minor comments for the authors:

In the submitted manuscript, figures were included as part of the Word document, resulting in multiple copies of the figures in my reviewer PDF. One of the figures in the Word document was an older version of Figure 2. The authors should remove the images from the manuscript document before their next submission.

Line 48: "analyses is that it because the the number" should read "analyses is that because the number"
Line 50: "quantitive" should read "quantitative"
Line 123: "ram" should read "RAM"
Line 124: Remove second trailing period from the sentence.
Line 132: The citation for Bowtie2 is incorrect; the provided citation is for the Lighter paper.
Line 247: "RecA, RpoB" should be "recA, rpoB" to match usage throughout paper.

---

## Round 0.4 · accepted · Accept

· Academic Editor

Accept

The authors have adequately responded to the comments raised by the reviewers. The manuscript changes have been adequately incorporated.